# Physical, Chemical, and Mechanical Properties of Six Bamboo from Sumatera Island Indonesia and Its Potential Applications for Composite Materials

**DOI:** 10.3390/polym14224868

**Published:** 2022-11-11

**Authors:** Rudi Hartono, Apri Heri Iswanto, Trisna Priadi, Evalina Herawati, Farhan Farizky, Jajang Sutiawan, Ihak Sumardi

**Affiliations:** 1Department of Forest Products, Faculty of Forestry, Universitas Sumatera Utara, Medan 20155, Indonesia; 2Forest Product Department, Faculty of Forestry and Environment, IPB University, Bogor 16680, Indonesia; 3School of Life Sciences and Technology, Institut Teknologi Bandung, Bandung 40132, Indonesia

**Keywords:** basic properties, bamboo, composite, composites materials

## Abstract

The suitability of bamboo’s basic characteristics is very important for more specific purposes, such as composite raw materials. Anatomical, physical, mechanical, and chemical characteristics are some of bamboo’s fundamental characteristics. This study analyses the basic properties, such as physical, mechanical, and chemical properties of bamboo from the Forest Area with Special Purpose (FASP) Pondok Buluh Sumatera Island, Indonesia (I); analyses the relationship between the properties of each type of bamboo (II); and chooses the type of bamboo with the best properties that have the potential to be applied to composite materials, such as laminated bamboo (III). This study used materials consisting of six species of bamboo from the FASP Pondok Buluh. The manufacture of physical and mechanical test samples refers to the ISO 22157 standard, 2004, while the chemical properties test refers to the TAPPI 1999 standard. The chemical, physical, and mechanical properties of bamboo vary widely among species. The lowest holocellulose and α-cellulose content were found in the Kuning Bamboo (*B. vulgaris* var. *vittata*). The content of holocellulose and α-cellulose causes the lowest density in Kuning Bamboo (*B. vulgaris* var. *vittata*). The Dasar Bamboo (*Bambusa vulgaris*) has the highest levels of lignin. The substances have an impact on moisture content, T/R ratio, and shear strength. The Dasar Bamboo (*Bambusa vulgaris*) has the lowest moisture content, the highest T/R ratio, and the highest shear strength. However, Betung Bamboo (*Dendrocalamus asper*) has the highest density in this study. The compressive strength of the Betung Bamboo (*Dendrocalamus asper*) has the highest value. Therefore, Betung bamboo and Dasar Bamboo in this study were potentially utilized for composite materials, such as laminated bamboo.

## 1. Introduction

Bamboo is a plant used to make non-timber forest products belonging to the Poaceae family. Bamboo plants have cylindrical stems of segments and nodes [1]. Bamboo has a relatively quick growth cycle and can mature in 3–4 years [2] with a seasonally variable growing length of 30–100 cm [3]. Bamboo has the potential to replace wood because of its rapid cycle.

Indonesia is one of the countries with a high diversity of bamboo species after China and India [4]. The number of bamboo species identified in Indonesia reached 161 from 22 genera in 2014 [5]. Meanwhile, in 2019, the number of identified bamboo species increased to 176 from 25 genera [6]. In addition, Indonesia’s total bamboo production in 2020 reached 11.3 million stems [7], and in 2021 this number increased, with total production coming to 50.1 million [8].

Bamboo is utilized for a variety of purposes, including light and heavy construction [9], as a raw material substitution in pulp and paper [10], and as a component in medicines [11]. Because wide varieties of bamboo have comparable or superior physical and mechanical qualities to wood, they are frequently used in the building industry. Bamboo’s typically long fibers allow it to be used as a substitute for wood in the pulp and paper industry [12]. In addition, bamboo can be utilized as a medicine to manage diabetes [13]. Bamboo can be used as a raw material for art tools, furniture, and traditional crafts. Several studies have also shown the use of bamboo as a raw material for biomaterial and composite products, such as plywood, oriented strand boards, and laminated bamboo [14,15,16].

The suitability of bamboo’s basic characteristics is very important for more precise purposes, such as for composite raw materials. Anatomical, physical, mechanical, and chemical characteristics are some of bamboo’s fundamental characteristics. Bamboo’s morphological characteristics (fiber size) and chemical characteristics (α-cellulose and hemicellulose) are closely associated with the pulp and paper industry. Holocellulose and lignin are typically found in cell walls. The pulp and paper produced will depend on how much lignin and holocellulose are present. Fatriasari and Hermiati [17] state that lignin content impacts color and chemical use during cooking, whereas holocellulose affects pulp color and fiber elasticity. The age of the bamboo also affects the amounts of lignin and holocellulose. According to Wang et al. [18] and Sadiku et al. [10], various chemical elements, including lignin content, holocellulose, and ash content, have a substantial impact on the age of bamboo.

Chemical properties also correlate with the mechanical strength produced by bamboo. Jansiri et al. [19] stated that the lignin content in bamboo would impact the bamboo’s mechanical properties. Bahtiar et al. [20] added that α-cellulose has a more extended bond structure and is inversely proportional to hemicellulose, so a higher ratio of α-cellulose content will have a good effect on the tensile strength of bamboo.

The mechanical and physical properties have been the subject of numerous studies in the past. According to Huang et al. [21], bamboo will become denser as it grows from the bottom to the top of the stem. This is carried on by the increase in vascular bundles from the stem’s bottom to the top. The moisture content of bamboo stems will decrease from the bottom to the top, according to Adam and Jusoh [22]. This is because from the bottom to the top of the bamboo stem, less parenchyma, which has a stronger ability to bind water than the vascular bundle, is present. Physical characteristics like density and moisture content will impact the mechanical qualities.

There is a connection between mechanical and physical qualities. The physical and mechanical characteristics of five varieties of bamboo were studied by Abdullah et al. [23]. They demonstrated that when compared to other bamboo with densities ranging from 0.54 to 0.62, Teman bamboo, with a density of 0.76, had greater tensile strength and modulus of elasticity value. This demonstrates that several bamboo’s mechanical qualities are positively impacted by its density. Additionally, bamboo’s mechanical and physical characteristics are affected by its axial position. According to studies by Awalluddin et al. [24] and Huang et al. [21], the top part of bamboo stems has more compressive strength than the middle and bottom. This is impacted by vascular bundles, which are more frequently present near the stems’ ends and affect the moisture content and density of bamboo stems. The mechanical characteristics of bamboo stems are impacted by their density and moisture content [25].

Bamboo’s basic characteristics may vary depending on where it is grown [18,26]. One forest area in North Sumatera with a large variety of bamboo species is the Forest Area with Special Purpose (FASP) Pondok Buluh. This forest area is located in the Dolok Panribuan sub-district, Simalungun Regency, North Sumatera Province, Indonesia. Besides the large variety of bamboo grown in this area, there is still less research about the characteristic of bamboo from this area. This study analyses the basic properties, such as physical, mechanical, and chemical properties of bamboo from FASP Pondok Buluh Sumatera Island Indonesia (I); analyses the relationship between their properties of each type of bamboo (II); and choose the type of bamboo with the best properties that have the potential to be applied to composite materials, such as laminated bamboo (III).

## 2. Material and Methods

### 2.1. Sample Preparation

This study used materials consisting of six species of bamboo, such as Betung (*Dendrocalamus asper*), Peol (*Dendrocalamus giganteus*), Butar (*Gigantochloa apus*), Lemang (*Schizostachyum bracycladum*), Kuning (*Bambusa vulgaris* var. *vittata*) and Dasar (*Bambusa vulgaris*) from the Forest Area with Special Purpose (FASP) Pondok Buluh. This forest area covers approximately 1272.70 hectares in Simalungun Regency, North Sumatra. The six bamboos were chosen because they are one of the most dominant plants in FASP Pondok Buluh.

A total of 18 stem bamboo are divided into several subsections on its height: the bottom, middle, and top (Figure 1). Each part makes samples for physical, mechanical, and chemical properties tests. The manufacture of physical and mechanical test samples refers to the ISO 22157 standard, 2004 [27], while the chemical properties test refers to the TAPPI 1999 standard [28].

### 2.2. Determination of Chemical Properties

Bamboo stems at the bottom, middle, and top parts are then made into powder form. The powder is then filtered using a 40–60 mesh sieve. Testing chemical properties is referred to as TAPPI and for the determination of holocellulose using the Browning (1967) method [29].

The method for determining holocellulose content is referred to by Browning (1967) [29]. Extractive-free sample (2 g) was put into an Erlenmeyer, then 100 mL of distilled water, 1 g of NaClO_2_, and 1 mL of acetic acid were added. The sample was heated at 70–80 °C using a water bath. Every 1 h from the reaction time, 1 g of NaClO_2_ and 0.5 mL of acetic acid were added up to 4 times. The sample was filtered and washed using hot distilled water and 25 mL of 10% acetic acid. The sample was then rewashed with hot distilled water until it was acid-free. The sample was dried in an oven at 103 ± 2 °C until the weight was constant.

The method for determining α-cellulose content refers to Browning (1967) [29]. 1 g of holocellulose sample was put into an Erlenmeyer, and then 10 mL of 17.5% NaOH was added at a temperature of 20 °C. For each time interval of 5 min, 5 mL of 17.5% NaOH was added three times to a total volume of 25 mL of 17.5% NaOH. The samples were left for 30 min at 25 ± 0.2 °C. After that, 33 mL of distilled water was added to the sample and left for 60 min. The sample was filtered and rinsed with 100 mL of 8.3% NaOH. Rinse continued with hot distilled water. After that, the sample was rinsed again with 10% acetic acid, followed by hot distilled water until free of acid. Samples were dried in an oven at 103 ± 2 °C for 24 h until the constant weight condition occurred.

Determining lignin klason levels was conducted with modifications to the TAPPI 222 om 88 standard [30,31]. Extractive-free samples of 0.5 g were hydrolyzed with 5 mL of 72% sulfuric acid for 3 h at room temperature while stirring every 15 min. The solution is diluted to a concentration of sulfuric acid of 3%. Hydrolysis was continued at a concentration of 3% sulfuric acid at 121 °C for 30 min in an autoclave. Lignin was precipitated, filtered, and washed with hot distilled water until free from acid, then dried at 103 ± 2 °C to constant weight. Meanwhile, the solubility of bamboo was measured to estimate the levels of bamboo extractives. Solubility was analyzed according to the TAPPI standard. The solubility of the extractive in benzene-alcohol refers to the TAPPI T 204 om-88 standard [32].

### 2.3. Determination of Physical Properties

The measured physical properties of bamboo include moisture content, density, and shrinkage. The sample size for bamboo’s density and moisture content is 3 × 2 cm in length and width. The test sample’s thickness adjusts to the bamboo’s thickness (1–2 cm). Meanwhile, the test sample size for shrinkage is 4 × 2 cm × bamboo thickness.

The moisture content of bamboo was measured in air-dry conditions. Air-dry conditions are the condition of bamboo after being dried with a fan under environmental conditions for 14 days. The oven-dry condition at 103 ± 2 °C is also used to determine the moisture content. The moisture content test is calculated using Equation (1). Meanwhile, the measured bamboo density includes air dry density. Density is calculated using Equation (2). In addition, bamboo shrinkage was measured in the tangential and radial directions. The observed shrinkage conditions include air-dry to oven-dry conditions. Measurement of shrinkage is done by measuring the dimensions of air-dry conditions. Then re-measured, the dimensions after drying as well. The volumetric shrinkage is calculated using Equation (3). The form of the physical and mechanical samples in this study are presented in Table 1.


(1)
$$\mathrm{Moisture}\;\mathrm{content}\;\left(\%\right)=\frac{ADW-ODW}{ODW}\times 100$$



(2)
$$\mathrm{Density}\left(g/{\mathrm{cm}}^{3}\right)=\frac{ADW}{ADV}$$



(3)
$$\mathrm{Shrinkage}\left(\%\right)=\frac{ADD-ODD}{ADD}\times 100$$


where:

*ADW* = Air-dry weight (g);

*ODW* = Oven-dry weight (g);

*ADV* = Air-dry volume (cm^3^);

*ADD* = Air-dry dimension (cm);

*ODD* = Oven-dry dimension (cm).

### 2.4. Determination of Mechanical Properties

The samples for compressive parallel to the grain, tensile parallel to the grain, and shear parallel to the grain were made according to ISO 22157-2; 2004. The sample’s height was equal to the bamboo diameter sample in the compressive strength sample. The compressive strength was obtained by applying a load to the sample from the vertical direction. Calculating the compressive strength can be conducted using Equation (4). Meanwhile, the sample of the tensile strength can be seen in Table 1. The calculation of the tensile strength can be carried out by Equation (5). In addition, the sample shear strength can be seen in Table 1. The shear strength value can be calculated using Equation (6).


(4)
$$\mathrm{Compressive}\;\mathrm{strength}=\frac{P}{A}$$



(5)
$$\mathrm{Tensile}\;\mathrm{strength}=\frac{P}{A}$$



(6)
$$\mathrm{Shear}\;\mathrm{strength}=\frac{P}{A}$$


where:

*P* = Maximum load (kg);

*A* = Cross-section area (cm^2^).

### 2.5. Statistical Analysis

This study used a 3 × 6 factorial experimental design with three replications. The factors studied included the part of the bamboo (bottom, middle, and top) and the type of bamboo (Betung, Peol, Butar, Lemang, Kuning, and Dasar). ANOVA analysis at a 95% confidence interval was carried out to determine the effect of the two single factors and the interaction between the two factors on the observed value. If the analysis results show significant results, a further test is carried out through the Duncan test to determine the factors that have a considerable effect.

### 2.6. Determination of the Best Bamboo

The bamboo scoring was made to determine the best bamboo. This scoring technique refers to Tarigan [33]. Bamboo scoring was made by involving physical and mechanical test parameters. The score values for the observation parameters density, T/R ratio, compressive strength, tensile strength, and Shear strength are divided into 6, with a detailed score of 1, for the parameter that has the lowest average value and a score of 6 for the parameter that has the highest average value. However, the moisture content observation parameter score is divided into 6 parts with a score of 1, for the details of the parameter with the highest value, and a score of 6 for the parameter with the lowest average value. The best bamboo is determined based on the highest total score.

## 3. Result and Discussion

### 3.1. Chemical Properties

Holocellulose is a polysaccharide fraction in wood cell walls consisting of cellulose and hemicellulose polymers [34]. Cellulose is the part that is resistant and insoluble in 17.5% NaOH, commonly called α-cellulose. Hemicellulose is soluble in 17.5% NaOH because it has a low polymerization chain compared to cellulose [35].

The holocellulose and α-cellulose content of six bamboo species from FASP Pondok Buluh ranged from 62.01–79.90% and 36.77–54.88% (Table 2). The highest amount of holocellulose was found in the bottom part of Lemang Bamboo, and the lowest amount was found in the top part of Kuning Bamboo. In contrast, the highest amount of α-cellulose was found in the bottom part of Dasar Bamboo, and the lowest amount was found in the bottom part of Kuning Bamboo. The cellulose and hemicellulose levels together affect the composite board’s quality [36,37].

A polymer of aromatic subunits produced from phenylpropane is called lignin. Several plant cell walls have lignin as a matrix around the polysaccharide components. This provides stiffness and compressive strength, making the walls hydrophobic and impermeable to water [38]. The examined bamboo lignin klason concentration ranged from 23.16 to 33.52%. (Table 2). The bamboo with the most lignin is the bottom part of Dasar Bamboo, whereas the bamboo with the least lignin is the middle part of Kuning bamboo.

Biomass contains non-structural chemical compositions known as extractive substances. Fengel and Wegener [39] defined extractives as non-cell wall components with diverse chemical compositions that function as food reserves and decay resistance. The six types of bamboo contain varying amounts of extractive substances, depending on the variety of bamboo. Generally, bamboo contains extractive materials such as resins, lipids, waxes, tannins, pentosan, hexosan, starch, and silica [17]. The content of extractive substances dissolved in ethanol-benzene ranged from 1.93–7.95% (Table 2). The bottom part of Peol Bamboo had the highest concentration of dissolved extractives in ethanol-benzene, whereas the middle part of Kuning bamboo had the lowest concentration. Resins, lipids, waxes, and tannins are among the materials that dissolve in the ethanol-benzene solution [39]. The concentrations of these extractives influence how much glue is used, how quickly the adhesive is curing, and how well the final composite board performs [40].

### 3.2. Physical Properties

The density of six species of bamboo is presented in Figure 2. The density varies between 0.60–0.83 g/cm^3^ with an average of 0.74 g/cm^3^. The statistical analysis showed an interaction effect between the height part of the bamboo and the type of bamboo on the density at the 95% confidence interval (Table 3). The highest density is found in the bottom part of Betung Bamboo at 0.83 g/cm^3^. Meanwhile, the lowest density was found in the bottom part of Kuning Bamboo at 0.60 g/cm^3^ (Table 4). This might result from Kuning Bamboo having less holocellulose and α-cellulose than another bamboo. According to Rowell [34], α-cellulose and holocellulose are the two primary components of cell walls. The cell wall’s thickness will depend on how much holocellulose and α-cellulose it has. Rowell [34] reported that cell walls affect the density of lignocellulosic materials.

Density is a unit of relatively constant value that may be used to assess bamboo uses. High-density bamboo must be cooked under more difficult nanocellulose separation conditions [41]. High-specific gravity raw materials are difficult to mill and fibrillate in the pulping process. Additionally, there will be an increase in the number of chemicals used [17]. Bamboo density depends on anatomical features like the proportion of vascular bundles and parenchyma [41]. According to Villareal et al. [42], the percentage of parenchyma that was more frequent at the bottom of bamboo was inversely related to the presence of vascular bundles, which were more frequently found at the top of bamboo stems than in other parts of the bamboo. Rusch et al. [43] added that the density is also influenced by the thickness of the bamboo cell wall, which will impact the bamboo’s use.

The results of this study showed that the average moisture content of bamboo was 122.80%. The moisture content of six bamboo species is presented in Figure 3. Statistical analysis showed that the interaction between bamboo parts and bamboo species significantly affected the moisture content of bamboo at a 95% confidence interval (Table 3). The moisture content of Dasar Bamboo at the top part shows the lowest moisture content of 62.31%, and Kuning Bamboo at the bottom part has the highest average moisture content of 223.4% (Table 4). This is probably affected by Kuning Bamboo’s lower lignin than other bamboo. Rowell [34] reported that lignin is a water-repellent chemical component. The proportion of parenchyma cells significantly impacts the moisture content of bamboo. Vascular bundles typically do not retain water as effectively as parenchyma cells do. At the top part of the bamboo stem, the proportion of vascular bundles was typically higher than at its bottom, where the proportion of parenchyma was higher. Compared to the top part of the bamboo stems, this results in high moisture content at the bottom [44]. Afterward, some of the mechanical strength of the bamboo will be impacted by the value of this moisture content.

The percentage of bamboo’s dimensional stability is based on how much it shrinks radially and tangentially. Tangential and radial shrinkage, sometimes known as the T/R Ratio, is a measure of dimensional stability. T/R ratios of bamboo that are close to or equal to 1 have consistent dimensions. Bamboo had a T/R ratio of 0.56 and 0.97 from air-dry to oven-dry conditions. According to the statistical analysis, bamboo shrinkage was significantly impacted by the interaction between bamboo parts and bamboo species at a 95% confidence level (Table 3). The T/R ratio of six bamboo species is presented in Figure 4. The top part of Lemang Bamboo has the lowest T/R ratio. Nevertheless, the bottom part of Dasar Bamboo has the highest T/R ratio (Table 4). This demonstrates that the bamboo from the bottom part of Butar bamboo has the greatest dimensional stability. This is possible because Dasar Bamboo has the highest lignin than another bamboo. According to Rowell [34], lignin is a water-repellent chemical substance. In general, the volume shrinkage of bamboo stems will depend on the presence of vascular bundles and parenchyma. There will be less volume shrinkage towards the top part of bamboo stems since there is less parenchyma than elsewhere [44]. The examined bamboo species revealed that the stem’s bottom asses the most shrinking. Additionally, multiple earlier research discovered the same outcome [10,45].

### 3.3. Mechanical Properties

The compressive strength in the bamboo ranged from 35 to 499 kg/cm^2^, averaging 217 kg/cm^2^. The compressive strength of six species of bamboo is presented in Figure 5. According to the statistical analysis, the compressive strength was significantly affected by the interaction between the part and the type of bamboo at a 95% confidence level (Table 5). The Betung Bamboo compressive strength at the top part has the highest value of 499 kg/cm^2^, while the top part of Kuning Bamboo has the lowest value of 35 kg/cm^2^ (Table 6). The Betung Bamboo’s higher density most apparently brings this on than other bamboo. Abdullah et al. [23] reported that density positively correlates with strength. The data on the average compressive strength obtained is lower than that of Nugroho and Bahtiar [46], which is 408–500 kg/cm^2^.

Physical characteristics and anatomical characteristics have an impact on bamboo’s mechanical strength. The presence of the vascular bundle affects the density and moisture content of the bamboo, which in turn affects its mechanical strength. According to Correal and Albelaez [47], bamboo’s density, which is impacted by the proportion of parenchyma and vascular bundle, will affect several mechanical properties, including compressive strength. According to Huang et al. [21], the bamboo stem’s top has the highest density, impacting the stem’s highest point of compressive strength. Except for bamboo Betung, the findings of studies on a variety of bamboo species, however, differed from those of various earlier investigations [21,47,48].

The shear strength test bamboo is 32–77 kg/cm² with an average of 55 kg/cm². The shear strength of six species of bamboo observed is presented in Figure 6. According to the statistical analysis, the shear strength parallel to the fiber was significantly affected by the interaction between the bamboo species and the bamboo’s portion at a 95% confidence level (Table 5). The top part of Butar Bamboo has the lowest shear strength at 32 kg/cm^2^, while the bottom part of Dasar Bamboo has the highest shear strength at 77 kg/cm^2^ (Table 6). Dasar Bamboo’s moisture content (low) most likely causes this compared to other bamboo. Abdullah et al. [23] reported that moisture content influences mechanical properties.

Similar to the compressive and tensile strength, the shear strength in bamboo is likewise affected by the increasing proportion of vascular bundle from the bottom to the top of the bamboo stem, making the mechanical strength at the top part of the bamboo greater than in other portions [21]. Even so, the results of several further research [47,49,50] that claimed the highest shear strength was discovered in the top part of the bamboo stem contradict a majority of the species of bamboo observed in this study.

The average tensile strength of bamboo in this study of 932–2868 kg/cm² with an average of 2123 kg/cm². The tensile strength of six bamboo species is presented in Figure 7. According to the statistical analysis, the tensile strength was significantly affected by the interaction between the part of the bamboo and the bamboo species at a 95% confidence level (Table 5). The bottom part of Betung bamboo has the highest tensile strength with a value of 2868 kg/cm^2^, while the middle part of Butar bamboo has the lowest value of 932 kg/cm^2^ (Table 6). The increase in vascular linkages to the top part of the stem is assumed to be the cause of bamboo’s tendency to become stronger as it rises from the bottom to the top of the stem.

Research on tensile strength has been conducted by several researchers before. [24,48,51]. These studies results demonstrated an inconsistent pattern of findings. Theoretically, physical characteristics such as moisture content and density can affect tensile strength, which means that the top part of the bamboo will typically have more tensile strength than the bottom and middle. However, the study by Rochim et al. [52] shows that bamboo’s highest tensile and compressive strength is not consistently located at the top part of the stem. According to Bahtiar et al. [20], the chemical characteristics of bamboo, such as lignin, cellulose, and hemicellulose, are also associated with the tensile strength of bamboo.

### 3.4. The Best Properties of Bamboo

The average value for bamboo properties is shown in Table 7. Meanwhile, the scoring results are shown in Table 8. The scoring results show that Betung and Dasar bamboo have the highest total scores. The relationship between the properties of bamboo can be seen in this study. The Betung bamboo in this study had the highest density (0.83 g/cm^3^), which is why the Betung bamboo in this study had the highest compressive strength (446 kg/cm^2^). Abdullah et al. [23] reported that several bamboo’s mechanical qualities are positively impacted by its density. In addition, the Dasar bamboo in this study has the highest lignin compared to other bamboo, which is why the basic bamboo has the best moisture content, T/R Ratio, and shear strength. Jansiri et al. [19] reported that the lignin content in bamboo would impact the bamboo’s physical and mechanical properties. Therefore, Betung bamboo and Dasar Bamboo in this study were potentially utilized for composite materials such as bamboo laminated.

## 4. Conclusions

The chemical, physical, and mechanical properties of bamboo vary widely among the species. The lowest holocellulose and α-cellulose content were found in the Kuning Bamboo (*B. vulgaris* var. *vittata*). The content of holocellulose and α-cellulose causes the lowest density in the Kuning Bamboo (*B. vulgaris var. vittata*). The Dasar Bamboo (*Bambusa vulgaris*) has the highest levels of lignin. The substances have an impact on moisture content, T/R Ratio, and shear strength. The Dasar Bamboo (*Bambusa vulgaris*) has the lowest moisture content, the highest T/R ratio, and the highest shear strength. This demonstrates that the Dasar bamboo has the greatest dimensional stability and good strength. In this study, density impacts the mechanical characteristics of bamboo. Betung Bamboo (*Dendrocalamus asper*) has the highest density in this study. The compressive strength of the Betung Bamboo (*Dendrocalamus asper*) has the highest value. Therefore, Betung bamboo and Dasar Bamboo in this study were potentially utilized for composite materials such as bamboo laminated.

## Figures and Tables

**Figure 1 polymers-14-04868-f001:**
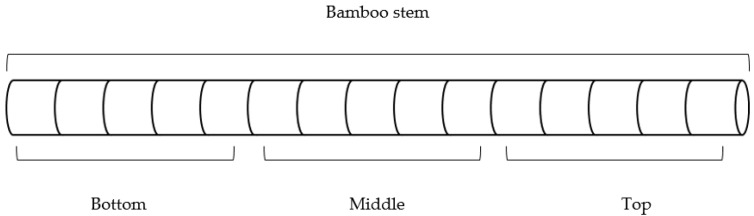
Subsection scheme of bamboo stem.

**Figure 2 polymers-14-04868-f002:**
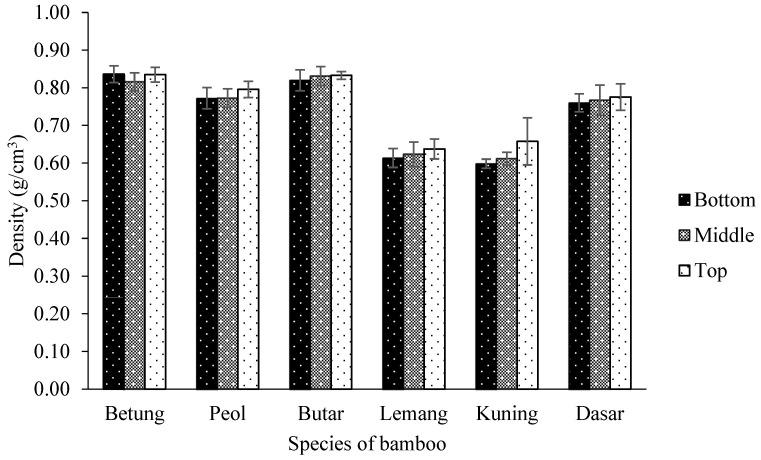
The density of six species of bamboo.

**Figure 3 polymers-14-04868-f003:**
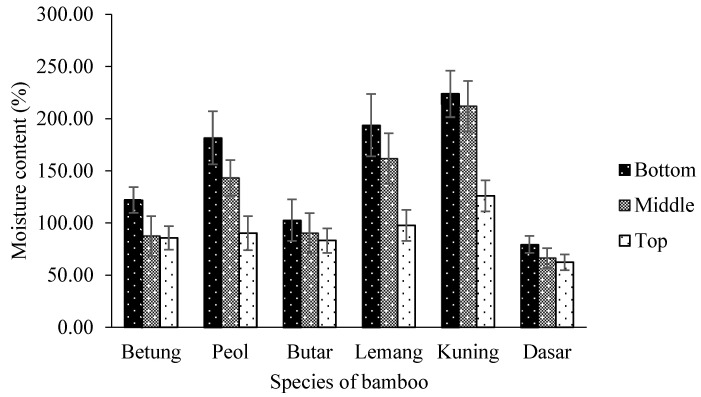
The moisture content of six bamboo species.

**Figure 4 polymers-14-04868-f004:**
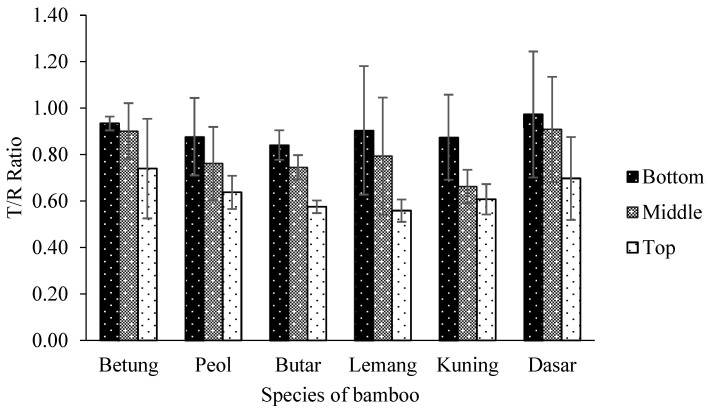
T/R ratio of six bamboo species.

**Figure 5 polymers-14-04868-f005:**
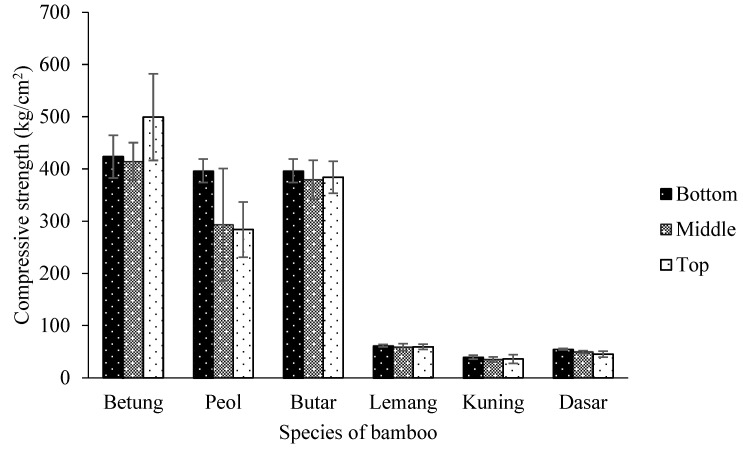
Compressive strength of six species of bamboo.

**Figure 6 polymers-14-04868-f006:**
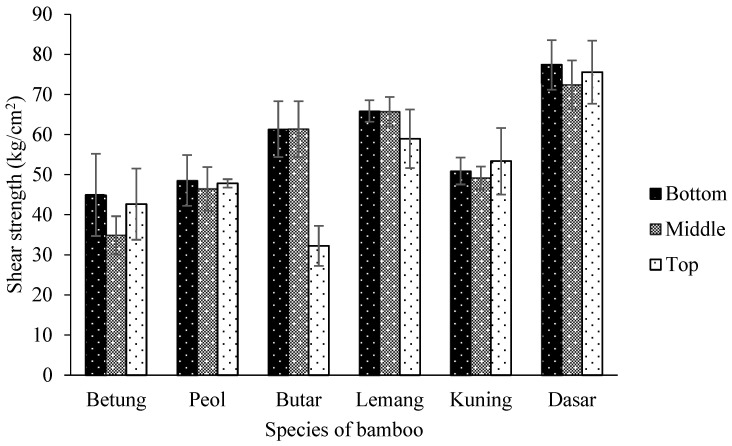
Shear strength of six species of bamboo.

**Figure 7 polymers-14-04868-f007:**
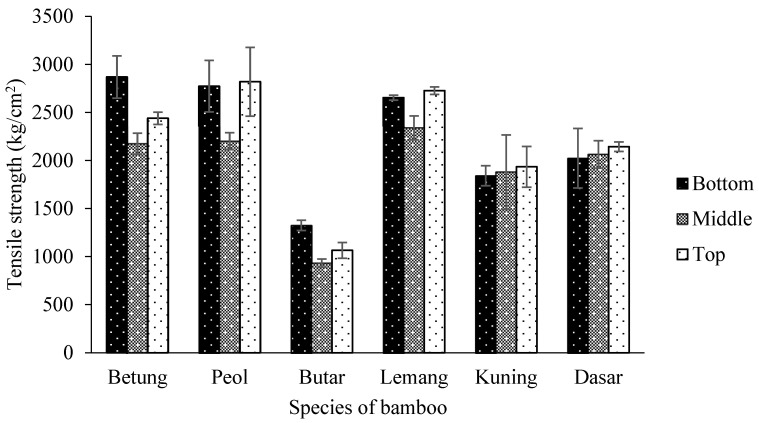
Tensile strength of six bamboo species.

**Table 1 polymers-14-04868-t001:** Physical and mechanical properties sample form.

No	Figure Sample	Testing
1	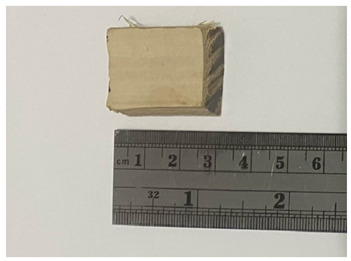	Density, moisture content, and T/R Ratio
2	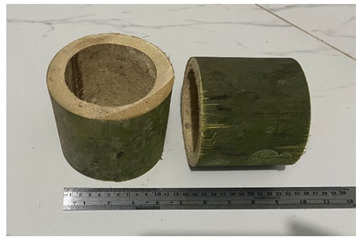	Compressive strength
3	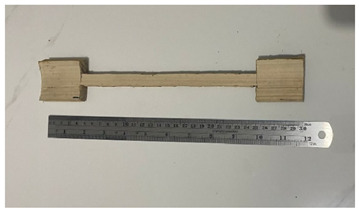	Tensile strength
4	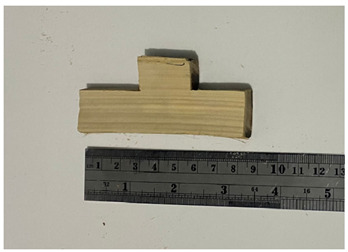	Shear strength

**Table 2 polymers-14-04868-t002:** The chemical component of six species of bamboo.

Species	Part	Extractive (%)	Holocellulose (%)	α-Cellulose (%)	Lignin Klason (%)
Betung (*Dendrocalamus asper*)	Bottom	4.27	71.99	48.59	28.96
	Middle	5.92	73.22	49.64	28.14
	Top	4.83	71.72	44.55	26.98
Peol (*Dendrocalamus giganteus*)	Bottom	7.95	72.58	52.36	31.78
	Middle	7.39	71.94	49.53	25.82
	Top	6.90	71.59	48.25	25.52
Butar (*Gigantochloa apus*)	Bottom	5.80	72.69	50.88	25.48
	Middle	5.90	71.96	49.29	27.34
	Top	5.26	71.52	49.24	26.94
Lemang (*Schizostachyum bracycladum*)	Bottom	2.60	79.90	50.14	30.44
	Middle	2.76	76.74	51.09	24.40
	Top	3.20	77.51	49.46	25.96
Kuning (*B. vulgaris* var. *vittata*)	Bottom	2.46	73.81	36.77	23.60
	Middle	1.93	68.87	37.70	23.16
	Top	2.53	62.01	37.91	26.22
Dasar (*Bambusa vulgaris*)	Bottom	4.67	77.16	54.88	33.52
	Middle	5.56	75.90	51.56	29.08
	Top	5.41	75.00	51.43	28.76

**Table 3 polymers-14-04868-t003:** Variance analysis summary of physical properties.

Parameter	ANOVA
Density	0.000 **
Moisture Content	0.000 **
T/R Ratio	0.040 **

^ns^ not significance, ** highly significance difference.

**Table 4 polymers-14-04868-t004:** Duncan multi-range-test of physical properties.

Properties	Species of Bamboo						
	Part	Betung	Peol	Butar	Lemang	Kuning	Dasar
Density (g/cm^3^)	Bottom	0.84 e	0.77 cd	0.82 de	0.61 ab	0.60 a	0.76 c
	Middle	0.82 de	0.77 cd	0.83 e	0.62 ab	0.61 ab	0.77 cd
	Top	0.83 e	0.80 cde	0.83 e	0.64 ab	0.66 b	0.78 cd
Moisture Content (%)	Bottom	122.12 def	181.77 hi	102.78 cde	193.85 ij	223.75 j	79.54 abc
	Middle	87.58 abc	143.22 fg	90.28 abcd	161.74 gh	211.96 ij	66.42 ab
	Top	85.78 abc	90.35 abcd	83.18 abc	97.76 bcde	126.07 ef	62.32 a
T/R Ratio	Bottom	0.93 de	0.88 bcde	0.84 abcde	0.90 cde	0.87 bcde	0.97 e
	Middle	0.90 cde	0.76 abcde	0.74 abcde	0.79 abcde	0.66 abcde	0.91 cde
	Top	0.74 abcde	0.64 abcd	0.58 ab	0.56 a	0.61 abc	0.70 abcde

Value with the same letter within a row is not significantly different.

**Table 5 polymers-14-04868-t005:** Variance analysis summary of mechanical properties.

Parameter	ANOVA
Compressive strength	0.000 **
Shear strength	0.000 **
Tensile strength	0.000 **

^ns^ not significance, ** highly significant difference.

**Table 6 polymers-14-04868-t006:** Duncan multi-range-test of mechanical properties.

Properties	Species of Bamboo						
	Part	Betung	Peol	Butar	Lemang	Kuning	Dasar
Compressive strength (kg/cm^2^)	Bottom	424 c	396 c	396 c	61 a	40 a	55 a
	Middle	414 c	293 b	379 c	59 a	35 a	50 a
	Top	499 d	284 b	384 c	59 a	36 a	45 a
Shear strength (kg/cm^2^)	Bottom	45 bc	49 cd	61 efg	66 fgh	51 cde	77 i
	Middle	35 ab	46 c	61 efg	66 fgh	49 cd	72 ghi
	Top	43 abc	48 cd	32 a	59 def	53 cde	76 hi
Tensile strength (kg/cm^2^)	Bottom	2868 i	2773 hi	1325 b	2654 ghi	1843 c	2025 cde
	Middle	2174 cdef	2200 def	932 a	2341 efg	1880 cd	2063 cde
	Top	2440 fgh	2820 i	1065 ab	2727 hi	1934 cd	2144 cdef

Value with the same letter within a row is not significantly different.

**Table 7 polymers-14-04868-t007:** Average properties of six bamboos.

Average Properties	Betung	Peol	Butar	Lemang	Kuning	Dasar
Extractive (%)	5.01	7.41	5.65	2.85	2.31	5.21
Holocellulose (%)	72.31	72.04	72.06	78.05	68.23	76.02
α-cellulose (%)	47.59	50.04	49.80	50.23	37.46	52.62
Lignin Klason (%)	28.03	27.71	26.59	26.93	24.33	30.45
Density (g/cm^3^)	0.829	0.780	0.828	0.625	0.623	0.768
Moisture Content (%)	98.492	138.446	92.078	151.117	187.257	69.424
T/R Ratio	0.858	0.759	0.720	0.752	0.715	0.859
Compressive strength (kg/cm^2^)	446	324	387	60	37	50
Shear strength (kg/cm^2^)	41	48	52	64	51	75
Tensile strength (kg/cm^2^)	2494	2598	1107	2574	1885	2077

**Table 8 polymers-14-04868-t008:** Scoring analysis properties of six bamboos.

Scoring Properties	Betung	Peol	Butar	Lemang	Kuning	Dasar
Density	6	4	5	2	1	3
Moisture Content	4	3	5	2	1	6
T/R Ratio	5	4	2	3	1	6
Compressive strength	6	4	5	3	1	2
Shear strength	1	2	4	5	3	6
Tensile strength	4	6	1	5	2	3
Total	26	23	22	20	9	26

## Data Availability

The data presented in this study are available on request from the corresponding author.

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
