# Peer review of "Physical, Chemical, and Mechanical Properties of Six Bamboo from Sumatera Island Indonesia and Its Potential Applications for Composite Materials"

_polymers, 2022, doi:10.3390/polym14224868_

Round 1

Reviewer 1 Report

This paper deals with the specifics of six bamboo species from a Special Purpose Forest Area Pondok Bulah (Indonesia). In experimental part the contents of basic chemical constituents (extractives, holocellulose, alpha- cellulose and lignin), physical properties (moisture content, density and volumetric shrinkage) and mechanical properties (compressive, tensile and shear strength) of various bamboo species were determined.

Although the topic of this paper is somewhat interesting, the manuscript lacks additional experiments to be conducted (like the determination of L/D ratio of bamboo fibers), so it can be considered a sole scientific paper. In its current form, the manuscript can only be published as a professional paper. This is due to the fact that only basic analysis was performed and that for explanations of obtained results and their relations only the data obtained by other authors was used. For instance, in multiple parts of the manuscript the Authors refer to the contents and the distribution of the parenchyma that in this case wasn’t determined. This resulted with the fact that some of the conclusions made are more of a presumption than a conclusion based on obtained results.  Also, the topic of this paper doesn’t fit the scope of the Special Issue of the Polymers journal titled “Smart Natural-Based Polymers”, but can however be submitted to be published in other Special Issues of the Polymers journal.  

As for the specific remarks, bamboo indeed is a very specific lignocellulosic natural material, but it is unclear what did the Authors mean when they say “Bamboo is a smart material from nature”, or “It is crucial to use bamboo for its intended purpose…”. The Authors state that the statistical analysis was performed, and the results are explained based on it, but there is no evidence (a table, or annotations on Figures 2 to 7) that would support this claim. Also, the manuscript is full of general conclusions and knowledge like “…density impacts the mechanical characteristics of bamboo” and those like “Betung bamboo in this study is potentially utilized for composite materials” without specifying the composite material type (e.g. particleboards, fiberboards, wood-plastic composites, multi-layer parquet floorings, bio -composites, etc.) for production of which it would be considered suitable.

Author Response

Thank you very much for your comment, suggestion, and recommendation. All changes in our manuscript we highlight in yellow.

Best Regard

Rudi Hartono

Reviewer 2 Report

Dear Authors,

your study is quite interesting, but it should be discussed deeper, so the readers can your conclusion in the development of new composite materials.

I think the aim of the study has to be more specific. What do you want to obtain? Which general knowledge is expected from your study?

Please summarize the aim in the Abstract.

Keywords have to be changed to ones other than in the title. So your work will be better searchable in scientific databases.

Several studies (page 2)? Please cite them.

2.1. Why these 6 species are selected? Please justify.

How many stems were used?

Crucial comment  for your conclusion:

Why do only high density and compressive strength decide which bamboo species is potentially utilized for composite materials?

Why study other quantities? If they have no meaning in this aspect?

I think there is a lack of discussion of every quantity of bamboo studied in the aspect of bamboo utilization. What is the effect of hemicell. and cellulose, moisture content?

You didn't conclude it.

Author Response

(The authors gave the same response as above.)

Round 2

Reviewer 1 Report

Dear Authors,

I have read the new version of your manuscript and find it to be much better than the original. Some (but not all!) of the specifics were addressed properly, and the overall quality of the manuscript is higher. However, as it is still somewhat of a basic research I find it quite hard to evaluate the manuscript as a sole scientific paper. I will give the recommendation for it to be published in its current form, but will let the Editor decide on article type.

Kind regards

Author Response

Dear Reviewer

Thank you very much for your correction, recommendation, and suggestion. All changes in our manuscript we highlight in yellow.

Best Regard

Rudi Hartono

Reviewer 2 Report

Dear Authors,

thank you for the response but Im still thinking the manuscript has to be improved, because:

The aim has three steps, and I think only the first has been achieved.

We can't see any relationship between properties. How these relationships are determined? What do they look like?

Choosing process of the type of bamboo with the best properties that have the potential to be applied to composite materials has to be clearly described in the method and the results of this process have to be presented and discussed. So the conclusion is the consequence of these descriptions. The choice can not be arbitrary. Everyone should be able to do it on the basis of your method and results.

I described above what was missing in the first version and in my opinion, is still missing.

Author Response

(The authors gave the same response as above.)

Round 3

Reviewer 2 Report

Dear Authors,

now your article makes sense.

Always make sure to document the achievement of the research aim.

Good luck!